Evaluation of predictive maintenance efficiency with the comparison of machine learning models in machining production process in brake industry

Aydın Can 1 canaydinn@gmail.com
http://orcid.org/0009-0004-7814-7077 Evrentuğ Burak 2
1 Department of Management Information System, Dokuz Eylül University , İzmir, İzmir Province , Turkey
2 Department of Computer Programming, Izmir University of Economics , İzmir, İzmir Province , Turkey
Ye Jun
Electronic publication date: 2025 Jul 16
Publication date: 2025
Volume: 11
Electronic Location ID: e2999
Received 2024 Oct 31; Accepted 2025 Jun 10
Copyright: © 2025 Aydın and Evrentuğ
Copyright year: 2025
Copyright holder: Aydın and Evrentuğ
License: This is an open access article distributed under the terms of the Creative Commons Attribution License, which permits unrestricted use, distribution, reproduction and adaptation in any medium and for any purpose provided that it is properly attributed. For attribution, the original author(s), title, publication source (PeerJ Computer Science) and either DOI or URL of the article must be cited.
License URL: https://creativecommons.org/licenses/by/4.0/

Keywords: Machine learning, Predictive maintenance, Fault diagnosis

Funding: The authors received no funding for this work.

==============================
Background

The utilization of technologies such as artificial intelligence (AI) and machine learning (ML) in industrial sectors has become a crucial requirement to enhance the efficiency and stability of production processes. Regular maintenance of machines and early detection of faults play a critical role in ensuring uninterrupted production and business continuity. Predictive maintenance practices, combined with sensors and data analysis methods, enable the collection, analysis, and transformation of machine-related data into meaningful insights. As a result, the anticipation of potential machine failures, the execution of planned maintenance activities, and the prevention of unexpected downtime become possible. These methods not only improve productivity in production processes but also contribute to reducing maintenance costs.

Methods

This study aims to predict machine faults using data analysis methods and enhance the accuracy performance of these predictions for an industrial company that produces braking components. Comprehensive examination and analysis of data were conducted to understand the symptoms and relationships of machine failures. ML classification methods were employed in the relevant study.

Results

Challenges such as the imbalance of class distributions in the dataset, the presence of missing and outlier values, and the high costs of necessary equipment and training pose significant barriers to implementation. Addressing these issues is critical for achieving effective predictive maintenance solutions. In order to achieve more accurate results, data splitting and k-fold cross-validation methods were applied during the learning and testing phases to overcome the imbalance problem in the dataset, undersampling techniques were applied, and outlier detection and normalization processes were used to improve data quality. The model performances, evaluated through accuracy, precision, recall, and F1-score, area under the curve (AUC), Cohen’s Matthew’s correlation coefficient (MCC) were compared. Hyperparameter optimization was also performed, resulting in significant improvements in model performance. This study contributes to the literature in terms of predictive maintenance application, classification, and data partitioning techniques. The findings highlight the importance of data preprocessing and advanced modeling techniques in predictive maintenance and emphasize how addressing data challenges can enhance the overall performance and reliability of ML models.

Introduction

Industry 4.0, with its emphasis on smart factories and advanced technologies such as artificial intelligence (AI) and Internet of Things (IoT), forms the foundation for implementing predictive maintenance in modern production systems (Zhong et al., 2023; Aldahiri, Alrashed & Hussain, 2021). The use of Industry 4.0 technologies offers many benefits, such as making industries more efficient, faster, flexible, and scalable. Besides, they aid in optimizing production processes, reducing errors in production lines and facilities, and decreasing energy and resource consumption, thereby enhancing environmental sustainability (Çınaroğlu, 2017). These benefits increase the competitive structure of the industry and allow for more effective meeting of the organization’s and customer’s needs. One critical area within Industry 4.0 is fault detection and predictive maintenance, where machine learning (ML) and data-driven techniques have proven indispensable (Doğan & Baloğlu, 2020). In this study, predictive maintenance is applied to computer numerical control (CNC) machines used in braking component production, focusing on utilizing ML methods to predict faults and enhance maintenance accuracy (Bogehoj, 2016). This aligns with the broader goals of Industry 4.0, where such data-driven maintenance strategies are essential for ensuring uninterrupted operations and minimizing downtime.

Fault detection and predictive maintenance focuses on preventing unexpected equipment failures through advanced data-driven techniques. One of the fault detection tools which is reactive maintenance involves intervening when a failure occurs, preventive maintenance aims to prevent failures through regular checks at specified intervals. However, these traditional approaches can lead to either costly unplanned downtime or excessive maintenance expenses. Predictive maintenance uses ML and big data analytics to predict failures, ensuring that maintenance is only carried out when necessary (Özkat, 2021; Sindhu, 2020; Thessen, 2016; Tong et al., 2023). This reduces unplanned downtime, optimises maintenance costs and significantly increases operational efficiency.

Sensor technologies play a critical role in this process. Data such as temperature, vibration, pressure, oil levels, etc. collected from machines in real time are analyzed and potential failures are detected before they occur. By processing sensor data, maintenance teams can make data-driven decisions instead of traditional manual controls, reducing both inaccurate predictions and ensuring production continuity.

Despite the increasing adoption of predictive maintenance strategies across industrial sectors, a review of the current literature reveals several methodological and practical limitations that hinder the broader applicability and effectiveness of such approaches (Bektaş, 2020; Bousdekis et al., 2019; Kane et al., 2022). To better understand these limitations and define the specific contributions of the present study, the following section critically examines prior research on predictive maintenance, with a particular focus on data characteristics, modeling practices, and system integration.

Gaps in the literature

Conspicuous by the lack of data diversity and widespread applicability, most studies use narrow datasets based on a single site or facility, leading to uncertainty of performance in new conditions (Abbas, Al-haideri & Bashikh, 2019; Gougam et al., 2024; Amihai et al., 2018; Borgi et al., 2017; Tambake et al., 2024; Paszkiewicz et al., 2023; Merkt, 2019). Most existing IoT prototypes and ML studies use only a single model (e.g., artificial neural networks (ANN) or simple regression) and lack comprehensive benchmark and hyperparameter optimization (Justus & Kanagachidambaresan, 2022; Kannadaguli, 2020; Gougam et al., 2024; Abbas, Al-haideri & Bashikh, 2019; Taş, 2018). Many studies use only training accuracy or a single hold-out split, while cross-validation, discrete test set and comprehensive metric comparisons are often neglected (Aydin & Guldamlasioglu, 2017; Abbas, Al-haideri & Bashikh, 2019; Selvaraj & Min, 2023; Aydın et al., 2021). The developed prognostic curves and classification results often remain in laboratory conditions and are not integrated into real-time systems (Amihai et al., 2018; Amruthnath & Gupta, 2018; Çekik & Turan, 2025; Soylemezoglu, Jagannathan & Saygin, 2010; Abu-Samah et al., 2015; Kannadaguli, 2020; Shearer, 2000). Most studies only report technical performance and do not provide infrastructure recommendations for the integration of results into decision support modules or similar platforms and active use by maintenance teams (Kasiviswanathan et al., 2024; Çekik & Turan, 2025; Selvaraj & Min, 2023). Existing studies mostly collect data using different IoT-based sensors and methods (Nikfar, Bitencourt & Mykoniatis, 2022; Soylu et al., 2022; Yeardley et al., 2022), but rarely present a unified, standardized protocol or end-to-end data flow infrastructure. There has been no comprehensive study focusing on changes in model performance results by comparing ML models with changing ratios of learning and test data in existing research. Following the research, the question “Can the efficiency of predictive maintenance applications be increased by using different ML algorithms based on sensor data?” was determined as the research question.

In this work, we collect 30,622 CNC sensor and operating system records in real-time with the MTConnect protocol and generate a balanced and meaningful feature set with undersampling and SelectKBest after extensive preprocessing steps such as missing data cleaning, outlier suppression, and categorical transformations. We maximize the generalizability and reliability of the models by benchmarking nine classical machine learning algorithms (decision tree (DT), naive Bayes (NB), k-nearest neighbors (KNN), support vector machine (SVM), Adaptive Boosting (AdaBoost), random forest (RF), Categorical Boosting (CatBoost), Extreme Gradient Boosting (XGBoost) and Light Gradient Boosting Machine (LightGBM)) and an artificial neural network with manual hyperparameter optimization, both with 80–20% holdout and 10-fold cross-validation. Finally, by streaming live data with MTConnect Agent and integrating the results into the decision support module, we not only provide a highly accurate classification, but also an end-to-end solution that supports real-time maintenance decisions.

Amihai et al. (2018) and Borgi et al. (2017) each worked with limited data from a single plant or 155 profiles, which posed a risk of generalizability. This study used a richer and more diverse database with 30,622 samples collected from different sensor types (temperature, oil levels, spindle speed data, axis load, etc.) and strengthened the data quality with outlier removal and feature selection steps.

In conclusion, this study highlights how ML, feature engineering and advanced learning paradigms can directly contribute to Industry 4.0 goals. By transforming the brake component production line into a data-driven maintenance environment, the study provides practical insights into the integration of these technologies into smart factories.

In conclusion, this study highlights how ML, feature engineering and advanced learning paradigms can directly contribute to Industry 4.0 goals. By transforming the brake component production line into a data-driven maintenance environment, the study provides practical insights into the integration of these technologies into smart factories.

Materials and Methods

In the “Methodology” section of the study, predictive maintenance was implemented using ML algorithms to predict the maintenance status of CNC machines used in braking component production at Ege Fren Sanayi ve Ticaret A.Ş. The study adopts a data-driven approach to identify repair needs and optimize maintenance activities, aiming to improve the accuracy of predictive models for detecting equipment failures.

The methodology outlines key steps in a data science process, as shown in Fig. 1. These include handling missing data through imputation or removal, noise elimination, converting categorical variables to numerical formats, feature selection, and splitting the dataset into training and testing subsets. To address class imbalance, undersampling methods were applied, and hyperparameter tuning was used to optimize model performance. Robustness was evaluated using holdout and cross-validation techniques, with multiple ML algorithms, including a deep learning-based multilayer perceptron (MLP), compared based on metrics like accuracy, precision, recall, F1-score, area under the curve (AUC) score, Cohen’s Kappa, and Matthew’s correlation coefficient (MCC). The best-performing model was selected, and findings were documented.

Figure 1 Overall proposed methodology.

The use of ML is justified by its ability to outperform traditional fault detection methods in handling complex, data-rich environments. CNC machines generate large volumes of sensor data, which ML models efficiently analyze to predict faults and enable data-driven maintenance decisions. This study applies ML techniques to improve the precision and reliability of fault detection processes in braking component production.

Understanding the business

The Mazak CNC machine is used for high-precision brake lifting part production. It operates through automated processes including design, material placement, and precise cutting, leveraging advanced motion systems.

The production begins with part design using CAD software and G-code programming. Raw materials are securely placed for machining, and the CNC machine follows programmed paths with precision, ensuring optimal speed, depth, and accuracy. After machining, quality inspections are conducted before shipment.

The machine is equipped with sensors that monitor critical parameters like oil and coolant levels, motor temperatures, and X, Y, Z positions, recording data in real-time for performance optimization.

Data collection

The MTConnect data collection process integrates multiple components to standardize and transmit data from Mazak CNC machines:

Mazatrol is the control system of Mazak CNC machines that creates processing programs, controls machine movements, and performs CNC operations. Adapter serves as an intermediary, connecting the Mazatrol control system to an MTConnect-compatible data collection system. MTConnect Agent receives data from the adapter, processes it into an MTConnect-compatible XML format, and facilitates data transfer to other systems. Client Software allows users, such as operators and managers, to access and monitor real-time MTConnect data. The data flow starts with Mazatrol operations transmitted via an API to the adapter using TCP. The adapter collects this data and forwards it to the MTConnect Agent, where it is converted into a standardized XML format. Finally, the data are transmitted to a user-accessible environment via HTTP. The MTConnect data transfer process is shown in Fig. 2.

Figure 2 MtConnect data processing procedure.

Data set

Data recording from Mazak CNC machines occurs from two different areas. One method consists of data automatically generated from the operating system. The other data recording method involves sensors connected to the device. Table 1 lists the variables associated with the operating system and their explanations.

Table 1 Operating system dependent variables.

Feature name	Description	
Grease oil level	The amount of grease in the lubrication system of the equipment or machine	
Datetime	The time the data is saved.	
x axis loading	Measurement value of the mechanical load on the horizontal axis of the machine.	
y axis loading	Measurement value of the mechanical load on the vertical axis of the machine.	
z axis loading	Measurement value of the mechanical load on the reciprocating axis of the machine.	
Spindle loading	It is the mechanical load measurement value on the rotating shaft of the machine.	
Spindle speed	It refers to the rotation speed on the rotating shaft of the machine.	
Spindle temperature	It refers to the temperature value on the rotating shaft of the machine.	
Spindle temperature status	It refers to the overheating condition on the rotating shaft of the machine.	
Servo status	Refers to the functionality of the current state of the servo motor.	
Spindle condition	Spindle motor refers to the functionality of the current state of the component.	
Emergency status	It expresses the value of pressing the emergency stop button on the machine.	
System status	It indicates the operating status of the system with instant data.	

Table 2 shows the variables recorded by the sensors and their descriptions. The relevant sensor parts consist of level measurement and temperature sensors. These sensors were installed in the device by the company.

Table 2 Variables depending on sensors.

Feature name	Description	
Anamil coolant oil level	It refers to the oil level connected to the cooling system of the main shaft that enables the engine to run.	
Anamil coolant oil temperature	It refers to the oil temperature connected to the cooling system of the main shaft that enables the engine to run.	
Apparatus hydraulic oil level	It expresses the oil level used in the hydraulic working system of the machine apparatus.	
Apparatus hydraulic oil temperature	It refers to the oil temperature used in the hydraulic working system of the machine apparatus.	
Electric panel temperature	It refers to the internal temperature of the transformer unit connected to the machine.	
Hydraulic oil level	Refers to the oil level in the overall hydraulic system of the machine.	
Hydraulic oil temperature	It refers to the oil temperature in the general hydraulic system of the machine.	
M51 motor temperature	Refers to the temperature of the machine engine.	
Bench transformer temperature	It refers to the temperature of the transformer connected to the bench.	

The data collected through MTConnect is used in comma-seperated values (CSV) format. The dataset consists of 22 variables, including 16 continuous variables, six categorical variables, one time variable and 30,622 records in the CSV document created. Continuous variables represent sensor readings such as grease oil level, spindle speed, and electric panel temperature. Categorical variables include descriptors such as System Status (the target variable), Spindle Condition, and Emergency Status. The time variable, DateTime, provides timestamps for each observation. This data is instrumental in capturing real-time operational details like temperature, oil levels, load conditions on axes, and overall machine status. The dataset records 18 distinct system states under the system status variable, which reflect the operational conditions of the machinery. These states are categorized into two main groups: normal state and faults or warnings. The normal state accounts for 16,201 observations, representing approximately 65% of the dataset, and indicates the machine is functioning without any issues. Faults and warnings are categorized based on their frequency and type. Frequent faults include Warning: Memory Protection, observed in 193 records (~0.78%), and Warning: Invalid Format, appearing in 176 records (~0.71%). Rare faults, such as Fault: Overload, Warning: Soft Limit, and Warning: Collision Warning, occur in fewer than 10 observations each. Fault types are further classified into overloads, memory-related faults, invalid format errors, and emergency status warnings. The relevant records consist of data between 27.02.2023 and 05.05.2023. As can be seen in Table 3, there are two types of data structures in the data set: categorical and numerical. Table 4 also shows the number of filled values in each data set.

Table 3 Data set variable types and count of filled data.

Feature name	Count of filled data	Data type	
Grease oil level	25,317	Float	
DateTime	30,622	Object	
X axis loading	21,959	Float	
Y axis loading	21,958	Float	
Z axis loading	21,958	Float	
Spindle loading	21,956	Float	
Spindle speed	21,956	Float	
Spindle temperature	21,885	Float	
Spindle temperature status	21,365	Object	
Servo status	21,365	Object	
Spindle condition	21,365	Object	
Emergency status	21,357	Object	
System status	21,956	Object	
Anamil coolant oil level	23,459	Float	
Anamil coolant oil temperature	23,460	Float	
Apparatus hydraulic oil level	25,317	Float	
Apparatus hydraulic oil temperature	25,317	Float	
Electric panel temperature	23,459	Float	
Hydraulic oil level	25,317	Float	
Hydraulic oil temperature	25,317	Float	
M51 motor temperature	25,317	Float	
Bench transformer temperature	25,317	Float	

Table 4 Frequency values of target variable.

Values of target variable	Frequency	
Normal	21,413	
Warning: Memory protection	193	
Warning: Invalid format	176	
Warning: Memory protection (Auto Operation)	54	
Warning: No Mdi data	39	
Warning: Have the same program	23	
Fault: Cutting block start locked	12	
Warning: Cutting feed overload	9	
Warning: Invalid address entry	8	
Warning: Data cannot be renewed	7	
Warning: Automatic calculation impossible	6	
Fault: Overload	5	
Warning: Cursor position is wrong	5	
Warning: Mazatrol programming selected	2	
Warning: Collision	1	
Warning: Cannot change unit	1	
Warning: Data missing	1	
Warning: Soft limit	1	

The variable representing the system status is designated as the target variable and represents the main element that the model is trying to predict. This critical parameter describes the system’s current or future performance or state. All other variables are considered independent variables that influence the target variable.

Correlation analysis was conducted to examine the relationship between the variables, and the results are visualized in Fig. 3. Dark red indicates negative correlation, while dark blue represents a positive correlation, with darker the colors signifying stronger relationships. For example, a strong positive correlation is observed between M51 Motor Temperature and Bench Transformer Temperature, shown by a dark blue shade in the Fig. 3.

Figure 3 Correlation analysis.

The results of the correlation analysis between the numerical variables in the dataset are visualized in a heat map. Correlation coefficients range between −1 and 1; where values near 1 indicate a strong positive correlation, and values near −1 indicate strong negative correlation. Variables with a coefficient of 0 have no significant relationship.

The analysis revealsa moderate positive correlation between axis loading (e.g., X Axis Loading, Y Axis Loading) and Spindle Temperature and Spindle Speed (e.g., 0.64 between Y Axis Loading and Spindle Temperature). This indicates that an increase in axis loading can have an impact on spindle temperature and speed. However, low or non-significant correlations were found between some variables.

In particular, Anamil Coolant Oil Level, Apparatus Hydraulic Oil Level and Hydraulic Oil Level were left blank in the correlation matrix because these variables contain constant values and therefore their correlations cannot be calculated. It is foreseen that these fixed variables will make a limited contribution to the modeling process.

Following the correlation table indicating elevated values among most entities, a deeper analysis of the underlying causes and consequences of these findings was conducted. The correlation table derived from the dataset reveals that the correlation coefficients among the majority of items are 1.00, signifying perfect linear relationships. The outcome was further investigated to ascertain its underlying reasons.

Initially, variables including X Axis Loading, Y Axis Loading, and Z Axis Loading demonstrate substantial correlations owing to their restricted variability and analogous patterns across data. These variables probably denote interconnected physical processes or quantify overlapping phenomena, leading to markedly comparable behavior. Furthermore, it was noted that variables such as Spindle Loading and Spindle Speed exhibit a substantial correlation with axis loads, indicating a functional or deterministic relationship intrinsic to the examined system. This is anticipated in systems where mechanical or physical interdependencies link variables.

These relationships were also revealed by preprocessing. All missing values were removed during data purification, removing imputation artifacts. This method improved data integrity but may have increased predictable patterns by reducing noise and variability. Despite perfect correlations, similarity measures like cosine similarity may vary due to magnitude or distribution differences. Variables with similar trends but different magnitudes can have high correlation but low similarity. This shows that correlation measures linear relationships, while similarity indices measure other data patterns.

Data preprocessing

To ensure accurate and trustworthy ML models, the dataset was preprocessed to treat missing values, manage outliers, and resolve class imbalance, a major concern in this work.

As shown in Table 5, missing values occurred primarily due to internet outages during data recording. The Emergency Status variable had the highest number of missing entries. After identifying rows with missing values, the dataset was reduced from 30,622 to 19,472 records by removing 11,150 incomplete rows. Following this process, all columns were verified to contain no missing values.

Table 5 Missing data check.

Feature name	Empty value	
Grease oil level	5,305	
DateTime	0	
X axis loading	8,663	
Y axis loading	8,664	
Z axis loading	8,664	
Spindle loading	8,666	
Spindle speed	8,666	
Spindle temperature	8,737	
Spindle temperature status	9,257	
Servo status	9,257	
Spindle condition	9,257	
Emergency status	9,265	
System status	8,666	
Anamil coolant oil level	7,163	
Anamil coolant oil temperature	7,162	
Apparatus hydraulic oil level	5,305	
Apparatus hydraulic oil temperature	5,305	
Electric panel temperature	7,163	
Hydraulic oil level	5,305	
Hydraulic oil temperature	5,305	
M51 motor temperature	5,305	
Bench transformer temperature	5,305	

Certain variables, such as Grease Oil Level, Anamil Coolant Oil Level, Apparatus Hydraulic Oil Level, and Hydraulic Oil Level, consistently contained only zero values. This issue was communicated to the relevant company.

Outliers, which can distort model performance, were identified and corrected to ensure data reliability. Figure 4 illustrates the distributions of numerical variables prior to outlier processing.

Figure 4 Numerical variable histogram distributions.

For outlier detection, values outside the 0.05 and 0.95 range of each variable’s distribution were identified and adjusted to the nearest limit. This process effectively eliminated all outliers in the dataset.

The system state, used as the target variable, represent the class frequencies, constitutes the target variable in the data set used. The class frequencies, which are detailed in Table 4.

In line with the discussions with the company, normal values represent the optimal operating conditions of the machine, while other values indicate the need for maintenance. Two models were created: one for all system frequencies and another grouping system status as 1 for normal values and 0 for others, using feature engineering.

Data normalization uniformizes information with different scales, reducing model accuracy issues in ML and statistical analysis. This study used min-max normalization to adjust data to 0–1. This method normalized all dataset values.

The dataset used in this study has a class imbalance, with 16,201 “Normal” observations and 8,648 “Fault” observations. This discrepancy could prejudice ML algorithms, lowering minority class performance. We randomly selected 8,648 “Normal” observations to equal the “Fault” class count using undersampling. This change ensured fair class representation, improving model learning. The goal variable was divided into two groups: “Normal” (1, optimal operation) and “Fault” (0, fault conditions). This binary classification optimized modeling by distinguishing normal and incorrect scenarios. The final balanced dataset has 8,648 observations per class after these modifications, providing a solid training and assessment base.

Feature selection was performed to eliminate unnecessary or ineffective variables and identify those that improve model performance. The feature selection process was completed by selecting five variables to be modeled with the SelectKBest method. Related variables are DateTime, Spindle Temperature, Animal Coolant Oil Temperature, M51 Motor Temperature, Bench Transformer Temperature.

Data preprocessing steps, including missing data analysis, normalization, and outlier managementwere applied to optimize the data set for ML algorithms. Various algorithm were used and compared during modeling, including DT, NB, kNN, SVM, Adaptive Boosting (AdaBoost), RF, CatBoost, XGBoost and LightGBM.

During the training and evaluation of the model, train-test split and k-fold cross-validation methods were used to improve model performance. Cross-validation played a critical role in evaluating the overall performance of the model and minimizing problems such as overfitting.

A two-stage approach was adopted for the target variable. In the first stage, the model treated the target variable was simplified into two categories: “Normal” values (1) and “Abnormal” values (0), redefining the problem as binary classification. This dual approach was used to evaluate classification challenges at different levels and determine the most effective strategy.

The examination of the dataset indicated a substantial class imbalance: 16,201 instances in the “Normal” class and 8,648 in the “Abnormal” class. To rectify this, undersampling was implemented, decreasing the “Normal” class to 8,648 observations to align with the “Abnormal” class count. This produced a balanced dataset including 8,648 observations per class, so ensuring rigorous training and assessment.

Results

Since the target variable is categorical, classification metrics such as accuracy, precision, recall and F1-score were used to evaluate model performance.

The discrimination approach was first applied to all class frequencies of the system state variable. The target variable was subsequently categorized into two groups: normal values and erroneous values. The performance of the model was subsequently assessed for both scenarios. The dataset was divided into 80% for training and 20% for testing in the separation procedure. Table 6 displays the performance metrics for all class frequencies, whereas Table 7 encapsulates the outcomes for the binary classification framework of the target variable.

Table 6 Achievement performances of the model evaluating whole class frequencies created using the separation method.

Algorithm	Accuracy	Precision	Recall	F1-score	
DT	0.906	0.907	0.909	0.908	
NB	0.882	0.882	0.900	0.896	
kNN	0.906	0.906	0.909	0.908	
SVM	0.885	0.885	0.900	0.890	
AdaBoost	0.904	0.903	0.910	0.907	
RF	0.906	0.907	0.909	0.908	
CatBoost	0.901	0.901	0.910	0.906	
XGBoost	0.902	0.901	0.915	0.906	
LightGBM	0.904	0.906	0.907	0.907	
MLP	0.902	0.903	0.905	0.904	

Table 7 Achievement performance of the model evaluating the binary class frequencies formed using the separation method.

Algorithm	Accuracy	Precision	Recall	F1-score	
DT	0.950	0.950	0.950	0.950	
NB	0.922	0.922	0.950	0.936	
kNN	0.946	0.946	0.949	0.948	
SVM	0.925	0.925	0.950	0.937	
AdaBoost	0.944	0.945	0.950	0.947	
RF	0.946	0.947	0.949	0.948	
CatBoost	0.941	0.941	0.950	0.946	
XGBoost	0.947	0.946	0.950	0.948	
LightGBM	0.944	0.946	0.947	0.947	
MLP	0.945	0.944	0.949	0.946	

In the k-layer cross-validation method, k was set as 10 and model building was performed. Due to the dataset’s imbalance, partitioning could not be applied using all class values of the target variable. As a result, this method was only used with the binary structure of the target variable, which included normal and erroneous conditions. Table 8 presents the performance metrics for the binary classification structure.

Table 8 Model success performances evaluating binary class frequencies generated using K-layer cross-validation method.

Algorithm	Accuracy	Precision	Recall	F1-score	
DT	0.655	0.993	0.651	0.767	
NB	0.968	0.972	0.996	0.984	
kNN	0.663	0.990	0.659	0.772	
SVM	0.972	0.972	1.0	0.980	
AdaBoost	0.912	0.992	0.917	0.937	
RF	0.732	0.993	0.722	0.814	
CatBoost	0.853	0.993	0.855	0.910	
XGBoost	0.740	0.993	0.738	0.825	
LightGBM	0.817	0.989	0.821	0.888	
MLP	0.958	0.960	0.965	0.962	

Hyperparameters were utilized during the model building process to enhance the learning and testing capabilities of the algorithms. Table 9 lists the algorithms and the manually tuned hyperparameters used to achieve optimal performance.

Table 9 Hyperparameters used in models.

Algorithm	Hyperparameters	
DT	Criterion: gini, entropy, log_loss	
NB	No changes were made as there was no need for hyperparameters.	
kNN	N_neighbors: 2, 5, 10, 100,
Weights: uniform, distance,
Algorithm: ball_tree, kd_tree, brute,
Metric: euclidean, manhattan, chebyshev, minkowski,
P: 1, 2	
SVM	C: 1,
Kernel: linear, poly, rbf, sigmoid,
Degree: 3, 5	
AdaBoost	Random_state : 1	
RF	N_estimators: 10, 50, 100,
Critesion: gini, entropy, log_loss,
Max_features: sqrt, log2	
CatBoost	Learning_rate: 0.03, 0.1,
Depth: 4, 6, 8,
12_leaf_reg: 1, 3, 5, 7, 9	
XGBoost	Min_child_weight: 1, 5, 10,
Gamma: 0.5, 1, 1.5, 2, 5,
Subsample: 0.6, 0.8, 1.0,
Colsample_bytree: 0.6, 0.8, 1.0,
Max_depth: 3, 4, 5	
LightGBM	Num_leaves: 20, 40, 60, 80, 100,
Min_child_samples: 5, 10, 15,
Max_depth: −1, 5, 10, 20,
Learning_rate: 0.05, 0.1, 0.1,
Reg_alpha: 0, 0.01, 0.03	
MLP	Hidden_layer_sizes: (100),
Activation: relu,
Solver: adam,
Learning_rate_init: 0.01	

The hyperparameters shown in Table 9 were applied to the relevant algorithms and the model performance values are shown in Table 10.

Table 10 Performance of models built using hyperparameters.

Algorithm	Accuracy	Precision	Recall	F1-score	
DT	0.997	0.997	0.999	0.999	
NB	0.998	0.998	0.999	0.998	
kNN	0.995	0.996	0.998	0.997	
SVM	0.995	0.996	0.996	0.996	
AdaBoost	0.993	0.992	1.0	0.996	
RF	0.993	0.995	0.997	0.996	
CatBoost	0.991	0.995	0.996	0.996	
XGBoost	0.992	0.995	0.991	0.993	
LightGBM	0.999	0.999	1.0	0.998	
MLP	0.985	0.986	0.984	0.985	

Table 11 presents the confusion matrices of the models on the dataset before the undersampling process was applied.

Table 11 Confusion matrix results of all models built using hyperparameters.

Model	Train confusion matrix	Test confusion matrix	
AdaBoost	[[6627, 287], [58, 12907]]	[[1662, 72], [19, 3217]]	
Decision tree	[[6903, 11], [7, 12958]]	[[1729, 5], [14, 3222]]	
K-Neighbors	[[6914, 0], [0, 12965]]	[[1705, 29], [22, 3214]]	
Random forest	[[6900, 14], [4, 12961]]	[[1727, 7], [5, 3231]]	
Naive Bayes	[[6478, 436], [73, 12892]]	[[1620, 114], [21, 3215]]	
Gradient boosting	[[6839, 75], [7, 12958]]	[[1711, 23], [1, 3235]]	
LGBMClassifier	[[6905, 9], [7, 12958]]	[[1730, 4], [1, 3235]]	
XGBClassifier	[[6839, 75], [7, 12958]]	[[1711, 23], [1, 3235]]	
CatBoost	[[6839, 75], [7, 12958]]	[[1711, 23], [1, 3235]]	
MLP	[[6875, 39], [12, 12953]]	[[1712, 22], [10, 3225]]	

The performance metrics following the undersampling process are shown in Table 12.

Table 12 Performance values after undersampling.

Algorithm	Accuracy	Precision	Recall	F1-score	
DT	0.875	0.874	0.874	0.874	
NB	0.887	0.886	0.899	0.891	
kNN	0.884	0.882	0.885	0.883	
SVM	0.881	0.881	0.879	0.879	
AdaBoost	0.875	0.895	0.910	0.902	
RF	0.875	0.875	0.872	0.873	
CatBoost	0.871	0.875	0.875	0.875	
XGBoost	0.874	0.875	0.874	0.874	
LightGBM	0.901	0.905	0.912	0.908	
MLP	0.887	0.890	0.885	0.887	

Subsequent to the undersampling procedure, diverse hyperparameters were implemented across all algorithms, and their performance metrics were assessed. The hyperparameters have been implemented in the algorithms, and the ideal values have been identified to maximize performance for each model.

The hyperparameters for the AdaBoost model consist of param_n_estimators, denoting the quantity of decision trees, and param_learning_rate, which regulates the learning rate. The optimal outcomes for this model were attained with param_n_estimators configured at 200 and param_learning_rate established at 1.0.

The essential hyperparameters for the DT model include param_criterion, which specifies the splitting criterion; param_max_depth, which establishes the maximum depth of the tree; param_min_samples_split, which denotes the minimum number of samples necessary to split a node; and param_min_samples_leaf, which indicates the minimum number of samples required in a leaf node. The optimal outcomes for this model were attained with param_criterion configured to entropy, param_max_depth established at 20, param_min_samples_split set to 10, and param_min_samples_leaf defined as 1.

The param_var_smoothing hyperparameter in the Naive Bayes algorithm modifies the variance smoothing value. The optimal outcomes were attained with param_var_smoothing configured to 0.1.

The LightGBM model’s main hyperparameters are param_num_leaves, which sets the maximum number of leaf nodes, param_learning_rate, which controls learning, param_n_estimators, which specifies the number of decision trees, param_max_depth, which sets tree depth, and param_min_child_samples, which sets leaf node sample depth. With param_num_leaves set to 10.0, learning_rate to 0.1, n_estimators to 50.0, max_depth to −1.0 (meaning infinite depth), and min_child_samples to 10.0, this model performed best.

The CatBoost algorithm’s key hyperparameters are param_iterations, which controls the number of trees, param_learning_rate, which controls learning rate, param_depth, which controls tree depth, and param_l2_leaf_reg, which controls leaf value L2 regularization. The best parameters for this model were 200.0 iterations, 0.1 learning rate, 4.0 depth, and 3.0 l2 leaf reg.

The support vector machine (SVM) model’s main hyperparameters are param_C, which controls regularization strength, param_gamma, which controls training instance impact, and param_kernel, which determines the kernel function. Optimal results were attained with param_C configured to 0.1, param_gamma designated as Scale, and param_kernel specified as RBF.

The KNN algorithm has essential hyperparameters: param_n_neighbors, which determines the count of neighbors for classification; param_weights, which outlines the weighting approach for the neighbors; and param_metric, which indicates the distance measurement technique employed. The optimal outcomes for this model were achieved with param_n_neighbors configured to 3, param_weights designated as uniform, and param_metric specified as euclidean.

The essential hyperparameters for the Random Forest model are param_n_estimators, indicating the number of trees; param_max_depth, regulating the maximum depth of each tree; param_min_samples_split, defining the minimum number of samples necessary to split a node; param_min_samples_leaf, specifying the minimum number of samples in a leaf node; and param_criterion, determining the splitting criterion. The optimal outcomes for this model were achieved with param_n_estimators configured to 100, param_max_depth set to 10, param_min_samples_split established at 5, param_min_samples_leaf fixed at 1, and param_criterion designated as gini.

There were a number of hyperparameters utilized for MLP model. The hidden_layer_sizes parameter specifies the size and number of hidden layers in the network; the best setup was one layer with 100 neurons. The activation parameter specifies the activation function employed by neurons; the best performance was achieved with the Rectified Linear Unit (ReLU) function. The solver parameter determines the optimization algorithm employed during training; adaptive moment estimation (Adam) was chosen due to its effectiveness at dealing with sparse gradients. Finally, learning_rate_init specifies the initial learning rate, with 0.01 producing the most precise and stable convergence in this study.

Each method underwent testing with 10 distinct hyperparameter combinations, and Table 13 presents the accuracy, precision, recall, and F1 scores attained with the optimal hyperparameters.

Table 13 Best performance metrics across all algorithms.

Algorithm	Accuracy	Precision	Recall	F1-score	
AdaBoost	0.901662	0.856985	0.964426	0.907508	
Decision tree	0.936017	0.932504	0.940700	0.936201	
(KNN)	0.909939	0.918637	0.900588	0.909151	
Random forest	0.958524	0.935297	0.985770	0.959741	
Naive Bayes	0.707389	0.760575	0.604230	0.672788	
LightGBM	0.949049	0.922337	0.981036	0.950705	
CatBoost	0.954952	0.934732	0.978655	0.956103	
(SVM)	0.899152	0.874732	0.908655	0.896103	
MLP	0.922663	0.903454	0.944564	0.923766	

Although conventional performance measures like accuracy, precision, recall, and F1-score are typically used to measure the success of classification models, this research added other measures—AUC, Cohen’s Kappa, and MCC—to also determine the predictive capability of each model. Each of these metrics gives distinct information: AUC assesses the ability of the model to distinguish between positive and negative cases, Cohen’s Kappa adjusts for chance agreement, and MCC gives a balanced metric by considering all the components in the confusion matrix. As can be seen in Table 14, models such as random forest (AUC: 0.977), XGBoost (0.966), SVM (0.935), and MLP (0.954) performed well with high scores across AUC, Cohen’s Kappa, and MCC, indicating robust and reliable classification performance. Notably, Random Forest achieved the highest AUC score (0.977) followed by the others, all of which recorded relatively stable performance as reflective of applicability to real-life predictive maintenance applications.

Table 14 AUC, Cohen’s Kappa, and MCC scores after undersampling.

Model	AUC score	Cohen’s Kappa	MCC	
AdaBoost	0.943	0.750	0.750	
Decision Tree	0.874	0.749	0.750	
KNN	0.902	0.693	0.695	
Random Forest	0.977	0.842	0.842	
Naive Bayes	0.804	0.267	0.288	
SVM	0.959	0.791	0.792	
LGBM	0.84	0.815	0.801	
XGBoost	0.966	0.775	0.778	
CatBoost	0.834	0.801	0.806	
MLP	0.954	0.740	0.741	

The categorization, data balancing, and hyperparameter tuning methods showed significant model performance differences. Figure 5 compares binary vs multiclass classification, imbalanced vs balanced data, and standard vs optimized models.

Figure 5 Accuracy curves for different scenarios.

Binary vs multiclass classification

Binary classification distinguishes between “Normal” and “Others,” whereas multiclass classification encompasses all error classes. Training accuracy stabilizes about epoch 30, with validation accuracy soon thereafter, indicating that binary classification acquires knowledge rapidly and generalizes effectively. Multiclass classification initially exhibits lower accuracy and a more significant disparity between training and validation accuracy, suggesting generalization challenges associated with task complexity.

Imbalanced vs balanced data

Class imbalance significantly impacts model efficacy. The model rapidly memorizes the majority class in the presence of imbalanced data, resulting in diminished validation accuracy. Conversely, balanced data consistently enhances training and validation accuracy, hence boosting generalization and mitigating the limitations of unbalanced datasets.

Standard vs optimized model

Standard models with default hyperparameters exhibit overfitting and demonstrate sluggish learning, resulting in a disparity between training and validation accuracy. As the validation accuracy aligns with the training accuracy, improved models exhibit accelerated learning and enhanced generalization. This demonstrates how hyperparameter tuning enhances model performance.

Discussion and Conclusion

This study performed a comparative comparison of ML methods to improve the efficacy of predictive maintenance applications on CNC machines within the brake sector. The use of many algorithms, including AdaBoost, random forest, LightGBM, CatBoost, and support vector machine, together with precise performance measures for accuracy, precision, recall, and F1 score, has enhanced the results, offering a thorough review. The findings indicate that employing the undersampling technique to tackle class imbalance has markedly enhanced the attainment of more balanced and consistent results. Undersampling mitigated the predominance of majority class samples in the training dataset, enabling the algorithms to concentrate equally on both classes, thus enhancing overall performance, particularly in recall and F1 measures.

Of the evaluated models, random forest proved to be the most efficacious algorithm, attaining the highest accuracy of 95.85%, accompanied by significant precision, recall, and F1 scores of 93.52%, 98.58%, and 95.97%, respectively. This highlights the ensemble structure’s capacity to manage complex and varied datasets efficiently. Likewise, LightGBM and CatBoost achieved impressive results, with accuracies of 94.90% and 95.50%, respectively, demonstrating their efficacy in predictive maintenance applications. Although SVM exhibited commendable performance with an accuracy of 89.92%, its little inferior recall score (90.87%) suggests possible avenues for enhancement relative to ensemble-based models. These findings validate the effectiveness of ensemble methods and hyperparameter tuning in predictive maintenance procedures. The undersampling technique was essential in enabling these algorithms to properly manage unbalanced input, hence enhancing their reliability.

The model performances, evaluated by accuracy, precision, recall, and F1-score, were compared. Additionally, to ensure sound decision-making in high-stakes systems, this study emphasizes the use of enhanced evaluation metrics such as AUC, Cohen’s Kappa, and MCC, which offer a more informative statistical insight into classification goodness than traditional accuracy-based measures. The model performances were evaluated using a comprehensive set of metrics including accuracy, precision, recall, F1-score, AUC, Cohen’s Kappa, and MCC to ensure a robust comparison.

This study emphasizes the essential function of digitalization and decision support systems within operational contexts from a management information systems (MIS) viewpoint. Predictive maintenance applications, particularly in manufacturing industries, diminish downtime, improve operational efficiency, and minimize maintenance expenses by preemptively recognizing machine faults. The incorporation of ML algorithms enhances data-driven decision-making, allowing organizations to optimize resource allocation and improve strategic interventions. Undersampling exemplifies how data preparation techniques can be effectively employed to tackle real-world issues, such as imbalanced datasets in operational contexts.

Furthermore, the implementation of techniques such as hyperparameter optimization, undersampling, and k-fold cross-validation has been confirmed as effective methods to enhance the accuracy and dependability of systems employed in MIS. This study highlights the significance of utilizing ML models for predictive maintenance and diverse organizational activities, including supply chain optimization, financial analysis, and customer relationship management (CRM). These transdisciplinary applications establish a robust basis for data-driven change throughout business units.

This study demonstrates that ML technologies, particularly ensemble methods such as random forest, LightGBM, and CatBoost, are essential for improving organizational efficiency and decision-making. The remarkable effectiveness of ensemble models, together with the fair results achieved through undersampling, highlights the strategic ability of data science to align operational needs with managerial perspectives. These findings establish a robust foundation for future study and highlight the importance of integrating data science techniques with Management Information Systems to promote sustainable growth and competitive advantage in a digital economy. Addressing data imbalances using methods like undersampling ensures that ML solutions are equitable, robust, and highly effective across various business environments.

Supplemental Information

Supplemental Information 1 Code.

Supplemental Information 2 Raw Data.

Supplemental Information 3 Performance Metrics of the MLP models.

Supplemental Information 4 Literature Table.

Supplemental Information 5 Performance Metrics of the Adaboost Model.

Supplemental Information 6 Performance Metrics of the Decision Tree Model.

Supplemental Information 7 Performance Metrics of the K-Nearest Neighbors (KNN) Model.

Supplemental Information 8 Performance Metrics of the Random Forest Model.

Supplemental Information 9 Performance Metrics of the Naive Bayes Model.

Supplemental Information 10 Performance Metrics of the LightGBM Model.

Supplemental Information 11 Performance Metrics of the CatBoost Model.

Supplemental Information 12 Performance Metrics of the Support Vector Machine (SVM) Model.

Additional Information and Declarations

Competing Interests

The authors declare that they have no competing interests.

Author Contributions

Can Aydın conceived and designed the experiments, performed the computation work, authored or reviewed drafts of the article, and approved the final draft.

Burak Evrentuğ performed the experiments, analyzed the data, prepared figures and/or tables, and approved the final draft.

Data Availability

The following information was supplied regarding data availability:

The code and data are available in the Supplemental Files.

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
