# Peer review of "Evaluation of predictive maintenance efficiency with the comparison of machine learning models in machining production process in brake industry"

_PeerJ Computer Science, doi:10.7717/peerj-cs.2999_

## Round 0.1 · original submission · Major Revisions

According to the reviewers' comments and suggestions, this article needs major revisions.

Reviewer 1 ·

Basic reporting

This paper proposes Evaluation of Predictive Maintenance Efficiency with the Comparison of Machine Learning Models in Machining Production Process in Brake Industry. In general, this paper is well presented. The following issues can be further considered.

Experimental design

This paper proposes Evaluation of Predictive Maintenance Efficiency with the Comparison of Machine Learning Models in Machining Production Process in Brake Industry. In general, this paper is well presented. The following issues can be further considered.

Validity of the findings

This paper proposes Evaluation of Predictive Maintenance Efficiency with the Comparison of Machine Learning Models in Machining Production Process in Brake Industry. In general, this paper is well presented. The following issues can be further considered.

Additional comments

1. More background and motivation of this study can be added, in case the readers are not very familiar with the topic.

2. The descriptions of the well known knowledge can be properly reduced.

3. Why introducing machine learning method for the problem? What is the major benefits compared with traditional methods?

4. Some related works on this topic should be reviewed

5. A couple of ablation studies should be added to evaluate the effects of the key parameters of the proposed method on the performance.

·

Basic reporting

NA

Experimental design

NA

Validity of the findings

NA

Additional comments

The authors have presented a predictive maintenance strategy utilizing machine learning for the machining production process in brake systems. The study focuses on predicting machine faults using data-driven approaches to enhance performance in industrial applications. While the paper has several issues and shortcomings, the results hold significance in this field and are well-structured. The study falls within the scope of PeerJ Computer Science and can be considered for publication following an extensive major review.
Comments:
1. The template of the paper is substandard. Please ensure the text is evenly distributed within the margins.
2. Please include a description of the challenges in the abstract, followed by a clear highlight of your key contributions to this field.
3. The authors have mentioned that an unbalanced dataset is used in this study. What is the degree of imbalance across the datasets? Please elaborate on this and explain how the imbalance was addressed.
4. The citation style does not comply with the journal's requirements. Please review the latest papers and cross-check the author guidelines using this link.; https://peerj.com/about/author-instructions/cs
5. The introduction should also be revised to highlight the importance of machine learning, feature engineering, deep learning, and transfer learning for the smart factory and industry 4.0.
6. Please add some sentences of fault detection in the other application such as aircraft and other mobility systems as well.
7. Please reformate the paper style as there is no need of bullet points in the paper, it’s not a report see line 143, 144, 145, 146 and 147. Remove all the bullet points in the paper, check it throughout the whole manuscript.
8. What kind of data sets are used. And what kinds of various faults have been studied in this work. Please give a descriptive analysis of that.
9. Please add a figure that describes the overall proposed methodology of this work.
10. Please revise all the figures in the paper, the quality of the figures is too bad. Please revise it and make good quality figures.
11. Please fix the font of the paper, don’t use the upper caser letter only. Revise the tables.
12. Please make a table that includes the overall experimental procedure and parameters from this study.
13. Explain the correlation table, even for all the entities the values are 1.00, how is it possible? The correlation of each entity is 100% and still has a similarity index with other parameters as well. How is it possible?
14. Give the confusion matrix figures for all the cases.
15. Also draw the epoch and accuracy curves for various cases and explain it.
16. How the training data, testing and validation data sets are used for the fault detection. Please explain it in more detail.
17. Revise figure 1, it doesn’t give much information.

---

## Round 0.2 · Major Revisions

According to the reviewer's comments and suggestions, this article still needs major revisions.

Reviewer 1 ·

Basic reporting

This paper proposes Evaluation of Predictive Maintenance Efficiency with the Comparison of Machine Learning Models in Machining Production Process in Brake Industry. In general, this paper is well presented. The following issues can be further considered.

1. More background and motivation of this study can be added, in case the readers are not very familiar with the topic.

2. The descriptions of the well known knowledge can be properly reduced.

3. Why introducing the machine learning method for the problem? What is the major benefits compared with traditional methods?

4. Some related works on this topic should be reviewed, such as "Dynamic Vision-Based Machinery Fault Diagnosis With Cross-Modality Feature Alignment", "State of charge prediction of lithium-ion batteries for electric aircraft with swin transformer", "Data-driven deep learning approach for thrust prediction of solid rocket motors", etc.

5. A couple of ablation studies should be added to evaluate the effects of the key parameters of the proposed method on the performance.

Experimental design

This paper proposes Evaluation of Predictive Maintenance Efficiency with the Comparison of Machine Learning Models in Machining Production Process in Brake Industry. In general, this paper is well presented. The following issues can be further considered.

1. More background and motivation of this study can be added, in case the readers are not very familiar with the topic.

2. The descriptions of the well known knowledge can be properly reduced.

3. Why introducing the machine learning method for the problem? What is the major benefits compared with traditional methods?

4. Some related works on this topic should be reviewed, such as "Dynamic Vision-Based Machinery Fault Diagnosis With Cross-Modality Feature Alignment", "State of charge prediction of lithium-ion batteries for electric aircraft with swin transformer", "Data-driven deep learning approach for thrust prediction of solid rocket motors", etc.

5. A couple of ablation studies should be added to evaluate the effects of the key parameters of the proposed method on the performance.

Validity of the findings

This paper proposes Evaluation of Predictive Maintenance Efficiency with the Comparison of Machine Learning Models in Machining Production Process in Brake Industry. In general, this paper is well presented. The following issues can be further considered.

1. More background and motivation of this study can be added, in case the readers are not very familiar with the topic.

2. The descriptions of the well known knowledge can be properly reduced.

3. Why introducing the machine learning method for the problem? What is the major benefits compared with traditional methods?

4. Some related works on this topic should be reviewed, such as "Dynamic Vision-Based Machinery Fault Diagnosis With Cross-Modality Feature Alignment", "State of charge prediction of lithium-ion batteries for electric aircraft with swin transformer", "Data-driven deep learning approach for thrust prediction of solid rocket motors", etc.

5. A couple of ablation studies should be added to evaluate the effects of the key parameters of the proposed method on the performance.

---

## Round 0.3 · Major Revisions

Based on the reviewers' comments and suggestions, this revised article still needs major revisions.

Reviewer 1 ·

Basic reporting

This paper proposes Evaluation of Predictive Maintenance Efficiency with the Comparison of Machine Learning Models in Machining Production Process in Brake Industry. In general, this paper is well presented. The following issues can be further considered.

1. More background and motivation of this study can be added, in case the readers are not very familiar with the topic.

2. The descriptions of the well known knowledge can be properly reduced.

3. Why introducing the predictive maintenance method for the problem? What is the major benefits compared with traditional methods?

4. Some related works on this topic should be reviewed, such as "State of charge prediction of lithium-ion batteries for electric aircraft with swin transformer", "Data-driven deep learning approach for thrust prediction of solid rocket motors", "Dynamic Vision-Based Machinery Fault Diagnosis With Cross-Modality Feature Alignment", etc.

5. A couple of ablation studies should be added to evaluate the effects of the key parameters of the proposed method on the performance.

Experimental design

This paper proposes Evaluation of Predictive Maintenance Efficiency with the Comparison of Machine Learning Models in Machining Production Process in Brake Industry. In general, this paper is well presented. The following issues can be further considered.

1. More background and motivation of this study can be added, in case the readers are not very familiar with the topic.

2. The descriptions of the well known knowledge can be properly reduced.

3. Why introducing the predictive maintenance method for the problem? What is the major benefits compared with traditional methods?

4. Some related works on this topic should be reviewed, such as "State of charge prediction of lithium-ion batteries for electric aircraft with swin transformer", "Data-driven deep learning approach for thrust prediction of solid rocket motors", "Dynamic Vision-Based Machinery Fault Diagnosis With Cross-Modality Feature Alignment", etc.

5. A couple of ablation studies should be added to evaluate the effects of the key parameters of the proposed method on the performance.

Validity of the findings

This paper proposes Evaluation of Predictive Maintenance Efficiency with the Comparison of Machine Learning Models in Machining Production Process in Brake Industry. In general, this paper is well presented. The following issues can be further considered.

1. More background and motivation of this study can be added, in case the readers are not very familiar with the topic.

2. The descriptions of the well known knowledge can be properly reduced.

3. Why introducing the predictive maintenance method for the problem? What is the major benefits compared with traditional methods?

4. Some related works on this topic should be reviewed, such as "State of charge prediction of lithium-ion batteries for electric aircraft with swin transformer", "Data-driven deep learning approach for thrust prediction of solid rocket motors", "Dynamic Vision-Based Machinery Fault Diagnosis With Cross-Modality Feature Alignment", etc.

5. A couple of ablation studies should be added to evaluate the effects of the key parameters of the proposed method on the performance.

Reviewer 3 ·

Basic reporting

All comments have been added in detail to the last section.

Experimental design

All comments have been added in detail to the last section.

Validity of the findings

All comments have been added in detail to the last section.

Additional comments

Review Report for PeerJ Computer Science
(Evaluation of Predictive Maintenance Efficiency with the Comparison of Machine Learning Models in Machining Production Process in Brake Industry)

1. Within the scope of this study, this research seeks to enhance the accuracy and performance of predictive maintenance solutions by using data analysis methods and machine learning classification techniques to predict machine faults, while overcoming challenges such as data imbalance, missing values, and outliers in an industrial company.

2. Although the importance of the subject is mentioned at a certain level in the introduction, the difference of the study from the literature and its contributions to the literature should be stated in a more explanatory manner. In addition, a more detailed literature table should be included and this study should be highlighted more.

3. The dataset and amount used in the study and the data preprocessing steps are sufficient. The preprocessing of the dataset and the methods used increased the quality of the study.

4. As stated in Tables-6, 7 and 8, it is observed that nine different machine learning models were used in the study. When the literature is examined, it should be stated more clearly why these were preferred, while there are many different models related to this problem solution. In addition, adding a few additional models will increase the depth of the study.

5. Although a certain level of metrics are obtained in terms of evolution metrics; comment on the study in terms of metrics such as auc score, roc curve, cohen cappa and mcc score.

The study basically has a certain level and the potential to contribute to the literature. However, attention should be paid to the sections mentioned above.

---

## Round 0.4 · accepted · Accept

Based on the reviewers' comments, this article can be accepted for publication.

Reviewer 1 ·

Basic reporting

fine

Experimental design

fine

Validity of the findings

fine

Additional comments

fine

Reviewer 3 ·

Basic reporting

All comments can be found in the last section.

Experimental design

All comments can be found in the last section.

Validity of the findings

All comments can be found in the last section.

Additional comments

Review Report for PeerJ Computer Science
(Evaluation of Predictive Maintenance Efficiency with the Comparison of Machine Learning Models in Machining Production Process in Brake Industry)

The latest version of the paper and the responses given appear to be at a reasonable level.